# The Effects of the Spatial Extent on Modelling Giant Panda Distributions Using Ecological Niche Models

**Ziye Huang [1], Anmin Huang [1,*], Terence P. Dawson [2] and Li Cong [3]**

1    Department of Human Geography and Urban and Rural Planning, College of Tourism, Huaqiao University, Quanzhou 362021, China; ziye.huang@kcl.ac.uk

2    Department of Geography, King's College, London WC2R 2LS, UK; terry.dawson@kcl.ac.uk

3    Department of Tourism, College of Architecture Landscape, Beijing Forestry University, Beijing 100083, China; lisacong@bjfu.edu.cn

*    Correspondence: amhuang@hqu.edu.cn

**Abstract:** Climate change and biodiversity loss have become increasingly prominent in recent years. To evaluate these two issues, prediction models have been developed on the basis of ecological-niche (or climate-envelope) models. However, the spatial scale and extent of the underlying environmental data are known to affect results. To verify whether the difference in the modelled spatial extent will affect model results, this study uses the MaxEnt model to predict the suitability range of giant pandas in the Min Mountain System (MMS) area through modelling performed (1) at a nationwide scale and (2) at a restricted MMS extent. The results show that, firstly, both models performed well in terms of accuracy. Secondly, extending the modelling extent does help improve the modelling results when the distribution data is incomplete. Thirdly, when environmental information is insufficient, the qualitative analysis should be combined with quantitative analysis to ensure the accuracy and practicality of the research. Finally, when predicting a suitability distribution of giant pandas, the modelling results under different spatial extents can provide management agencies at the various administrative levels with more targeted giant panda protective measures.

**Keywords:** MaxEnt model; spatial extent; ecological niche; *Ailuropoda melanoleuca*; Min Mountain System

## 1. Introduction

Climate change and human activities have caused massive loss and fragmentation of animal and plant habitats and a sharp decline in the numbers of many rare species [1]. The wild giant panda (*Ailuropoda melanoleuca*) is one of the most globally endangered species, and its survival is uncertain. In 1869, the French priest Pere Armand David first discovered the giant panda in Sichuan province, China, and it attracted scholars' attention worldwide [2]. According to fossil evidence, documentary records, and quantitative surveys, the distribution of wild giant pandas has gradually shrunk; they have disappeared from most parts of China and from Vietnam, Laos, Myanmar, and other regions, and now remain only in parts of southwestern China and northwestern China [3,4]. Today, they are an endangered species unique to China and only distributed in six major Chinese mountain systems: the Qinling Mountain System, the Min Mountain System (MMS), the Qionglai Mountain System, the Daxiangling Mountain System, the Xiaoxiangling Mountain System, and the Liang Mountain System. These mountains are located in southwestern China, crossing Shaanxi, Gansu, and Sichuan provinces [5].

The Fourth National Giant Panda Survey Report (2015) reports the status of wild giant pandas in recent years [6]. The number of wild giant pandas increased by 268 individuals compared with the previous survey, reaching 1864 [7]. Habitats and potential habitats rose by 11.8% and 6.3%, respectively [6]. Even though the current survival situation of giant pandas has not continued to deteriorate based on data, the fragmentation of the habitat is

a hidden danger that needs continuous attention. The existing giant pandas are divided into 33 local populations, and there has been a decrease in connectivity between these populations and some of the habitats are of poor quality. Therefore, although the number of wild giant pandas has increased, the ecological isolation between local populations and poor quality of some habitats still poses a significant threat to the giant pandas' future survival [3,4]. According to Linderman et al. (2004), human influence in the next 30 years will lead to a 16% loss of giant panda habitat [8,9].

Therefore, it is essential to figure out what has caused this species to become endangered and to understand the factors that interfere with their natural environment to ensure the sustainable development of this endangered species; moreover, such issues need to be resolved urgently [10]. To ensure the suitability of the wild giant pandas' habitat and promote the formulation of protection measures, many scholars are devoted to studying the factors that give rise to the fragmentation of habitats and cause other threats to the survival of the population. The conclusion of these research results identify four categories of factors that primarily affect the giant pandas' survival and habitat fragmentation: (1) human activities, logging, reclamation, construction of cities and roads, grazing, and poaching; (2) ecological limitations of giant pandas, such as the special requirement for food and low reproduction rate [11]; (3) natural disturbances, such as earthquakes, and bamboo blossoming [12,13]; (4) climate change, the changes of plants' growth structure, temperature, rainfall, and other factors affected by climate warming.

However, due to the imbalance of scientific research structure, most scientific research work only focus on the aspect of phenomenon analysis, and relatively little work has been carried out from the perspective of sustainable development [10]. Presently, to increase the population and prevent the reduction of giant pandas' habitat, there is an urgent need for quantifying how different populations of giant pandas are predicted to respond to different climate change scenarios. At the same time, corresponding response measures should be made based on the model results to ensure the sustainable survival of giant pandas. Ecological niche models (ENMs) are currently used in geo-biological research, and are also the main modelling tool for predicting and solving sustainable ecological development [11]. Due to the practicality and accuracy of such models in solving ecological problems, ENMs have been widely used in environmental research, including issues such as climate change, the impact of species invasion on another species, and for predicting the potential distribution areas of different species [14,15]. The popular ENMs in recent years include: Genetic Algorithm for Rule-Set Prediction (GARP), Ecological-Niche Factor Analysis (ENFA), Bioclimate Analysis and Prediction System (BIOCLIM), and MaxEnt [16]. Of these, the MaxEnt model is the most widely used in studying the geographic-biological connection.

Few studies have examined the influence of spatial scale on the predictions generated by MaxEnt modelling. Of these, some studies have shown that different scales have minimal influence on model predictions [17], whereas other studies have concluded that variation in scale-dependent effects is influenced by environmental variable selection [5,18]. However, there remain few concrete recommendations on how to deal with these issues in order to improve MaxEnt modelling.

Based on the concern about the development of the giant pandas' living conditions and interest in further research on modelling extents, this study aims to examine how the spatial extent of the modelling affects the niche model result. Taking spatial extent as a breakthrough point to technically reduce the errors in the study of endangered species models and will make it possible to provide suggestions for future monitoring and protection methods for giant pandas and thereby ensure the survival of giant pandas and alleviate the conditions that threaten their survival. To accomplish this aim, the research has the following objectives: (1) to quantify some significant human disturbances and incorporate them into the environmental variables in the MaxEnt model; (2) to predict the accuracy of the model to ensure the reliability of follow-up research; (3) to model suitable habitats for giant pandas in different spatial ranges and identify some differences between them; (4) to

follow the results with further discussion and suggest how this research provides practical help for the future.

## 2. Materials and Methods

### 2.1. Introduction to Model

The MaxEnt model (Philips et al., 2006) takes the Maximum Entropy Theory proposed by Jaynes in 1957 as theoretical guidance [19]. It is a widely used method with high performance and is commonly used in climate prediction, biological protection, species invasion, and other biological research [5,20,21]. The principle of MaxEnt is that the 'dissipation' of the system increases the entropy. Until the species and the environment reach the maximum entropy, the environment will be in equilibrium [22]. The model predicts species' distribution by calculating the state parameters when the entropy is maximum [22]. The MaxEnt models exhibit little sensitivity or change to model accuracy with significant changes in data quantity, although processing times tend to be accelerated and model accuracy is higher when models are based on presence-only data [23]. Thus, follow-up research is based on the MaxEnt model with high accuracy, easy operation, and low data quantity requirements.

### 2.2. Case Study Area and Research Extent

Wild Chinese giant pandas are distributed in six major mountain systems of the Sichuan, Shaanxi, and Gansu provinces. The area and habitat ratio of each mountain system are shown in Table 1 [13]. The MMS is located in southwestern China, running through northern Sichuan province and southern Gansu province. The geographical position is $102° 70'–105.60°$ E and $31°40'–33.70'$ N, transitioning from the Yangtze River's upper reaches to the Tibetan Plateau [24]. The giant pandas in the MMS are mainly distributed in the mountainous dark coniferous forest belt at 2100 to 3400 m. The subalpine dark coniferous forest belt is at a height of 3000 to 3900 m in the alpine valley [13]. There are 27 established nature reserves in the MMS, where 44% of China's pandas are distributed [21].

**Table 1.** Statistics of the habitat area of giant pandas in each mountain system.

| Mountain Systems | Area/hm² | Percentage of Total Habitat/% | Administrative Regions/Province |
|---|---|---|---|
| The Qinling Mountain | 352,914 | 15.31 | Shaanxi, Gansu |
| The Min Mountain | 960,313 | 41.66 | Sichuan, Gansu |
| The Qionglai Mountain | 610,122 | 26.47 | Sichuan |
| The Daxiangling Mountain | 81,026 | 3.52 | Sichuan |
| The Xiaoxiangling Mountain | 80,204 | 3.48 | Sichuan |
| The Liang Mountain | 220,412 | 9.56 | Sichuan |
| Total | 2,304,991 | 100 | |

Source from: Zhang, Q., 2009. Study on Habitat Selection of Giant Panda in the Min Mountain, Gansu. Master. Northwest Normal University.

Since the MMS region is the most critical habitat for giant pandas among the six mountain systems, this area was selected as the target area for modelling. A national extent is selected as a control group, which contains all the known global distributional data for the giant panda. Comparing the model result under the MMS region and the nationwide extent (subsequently clipped to the MMS range) enabled us to evaluate variation in predicted suitable habitat areas. The specific modelling area comparison shows in Figure 1.

### 2.3. Data Collection

Species distribution data were derived from two sources; firstly the Global Biodiversity Information Faculty [25] and National Specimen Information Infrastructure [26]; and secondly, from all relevant peer-review literature records of giant pandas' occurrence. A total of 260 occurrence sites were obtained from these two sources. Google Earth was used to filter the distribution information to gain the relevant latitude and longitude coordinates,

as well as helping to remove 'double records' and locations with unknown geographic coordinates. This resulted in 89 occurrence records, of which 65 were from the MMS.

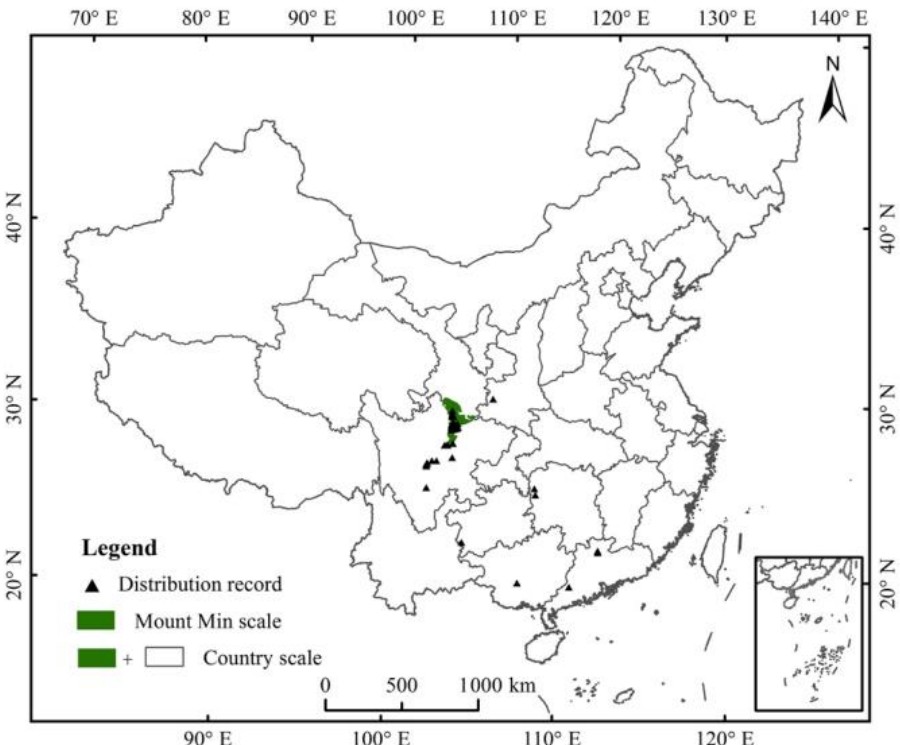

**Figure 1.** Modelling areas of spatial extent and distribution sites of the wild giant panda.

All climate and environmental data were obtained from WORLDCLIM [27]–a global database based on meteorological information recorded by weather stations worldwide from 1970 to 2000 (Table 2). It uses interpolation to generate global climate raster data with a spatial resolution of 30 arc-seconds (1 km$^2$) [17]. The roads, rivers, and residential areas data were derived from OpenStreetMap then calculated by the Euclidean distance tool in ArcGIS to acquire the distance to the residential area, the distance to the road, and the distance to the river, with resampling of 30 arc-seconds (1 km$^2$). The data of land use types in 2018 came from the Resource and Environment Data Center of the Chinese Academy of Sciences [28], with a resolution of 30 arc-seconds (1 km$^2$).

However, the redundancy and overfitting of variables can bias the data results [29]. To avoid overfitting, only when the environmental variables are reduced to a reasonable number can the accuracy and predictive ability of the model be improved [30,31]. Therefore, the environmental variable data values of 89 sample points were extracted by DIVA-GIS software. The Pearson correlation coefficient was used to test the multicollinearity between climate variables in a set of climate variables with high correlation (r > 0.8). According to the contribution rate, only one variable closely related to the species distribution or convenient for model interpretation was selected for model prediction. In this case, 11 environmental variables were ultimately screened at two extents, respectively, for model construction.

## 2.4. Model Construction

Firstly, giant pandas' distribution and environmental data were imported into the MaxEnt version 3.4.1 to model potential distributions of giant pandas at both spatial extents [32]. The maximum number of iterations (500 times) and the maximum number of background points (10,000) were kept recommended default settings. The cross-validation method was used to test the robustness of each model. The study used 25% distribution points as the test set and 75% distribution points as the training set and ran these 10 times repeatedly to make the data utilization rate higher [33]. Jackknife and percentage rate

were selected to evaluate the relative importance of each environmental factor on the giant pandas' distribution. The response curve was chosen to obtain specific values of environmental variables most suitable for the survival of giant pandas.

**Table 2.** 19 bioclimatic variables.

| Index | Description |
|---|---|
| Bio1 | Mean annual temperature |
| Bio2 | Mean diurnal air temperature range |
| Bio3 | Isothermality (Bio2/Bio7 $\times$ 100) |
| Bio4 | The standard deviation of temperature seasonality |
| Bio5 | Max temperature of the warmest month |
| Bio6 | Min. temperature of the coldest month |
| Bio7 | Temperature annual range (Bio5-Bio6) |
| Bio8 | Mean temperature of the wettest quarter |
| Bio9 | Mean temperature of the driest quarter |
| Bio10 | Mean temperature of the warmest quarter |
| Bio11 | Mean temperature of the coldest quarter |
| Bio12 | Annual precipitation |
| Bio13 | Precipitation of the wettest month |
| Bio14 | Precipitation of the driest month |
| Bio15 | Coefficient of variation of precipitation seasonality |
| Bio16 | Precipitation of the wettest quarter |
| Bio17 | Precipitation of the driest quarter |
| Bio18 | Precipitation of the warmest quarter |
| Bio19 | Precipitation of the coldest quarter |

The area under the curve (AUC) of receiver operating characteristics (ROC) was used to predict the model performance. When the ROC curve cannot indicate which classifier performs better, AUC provides a more intuitive way to predict the accuracy of models, which is a single measure of overall accuracy that does not depend on a particular threshold [34]. The value of AUC is between 0 and 1. The larger the AUC value, the better the prediction effect, and the higher the prediction accuracy. The general evaluation standard is less than 0.5 is model meaningless, 0.5–0.6 is poor, 0.6–0.7 is fair, 0.7–0.8 is more accurate, 0.8–0.9 is very accurate, and 0.9–1.0 is extremely accurate [35]. When the AUC value is more than 0.7, the prediction result of the MaxEnt model can be credible.

## 3. Results

### 3.1. Model Prediction Accuracy Test

The ROC curve was obtained after running the MaxEnt model 20 times repeatedly to ensure the model's prediction results' stability. AUC values of the MMS region (Figure 2) and the national extent (Figure 3) were 0.879 and 0.987, respectively, suggesting that the MaxEnt models for both scales performed well, and the national extent scaled model is more accurate than the MMS regional model.

### 3.2. Distribution of Predicted Suitable Areas for the Giant Panda

The suitable habitat distribution map under the MMS (Figure 4) and the country extent (Figure 5) was directly obtained through modelling. To make the information in the map of the national range clearer, it was clipped to the range of the MMS (Figure 6). There is a big difference in the suitability distribution map under the two areas modelling. Geographically, modelling at the region of the MMS, the highly suitable areas are only scattered in the core area of the MMS, such as the Wanglang Reserve, Hulu ditch, Wenxian ditch, and Xi ditch. However, when modelling on a national size, most of the MMS areas are suitable for wild giant pandas to survive, and at least half of the regions are highly suitable. In the whole country, only the MMS area has a highly suitable area. Except for the Gansu and Sichuan provinces covered by the MMS, only a few areas at the junction of Tibet and Yunnan provinces have some suitable regions.

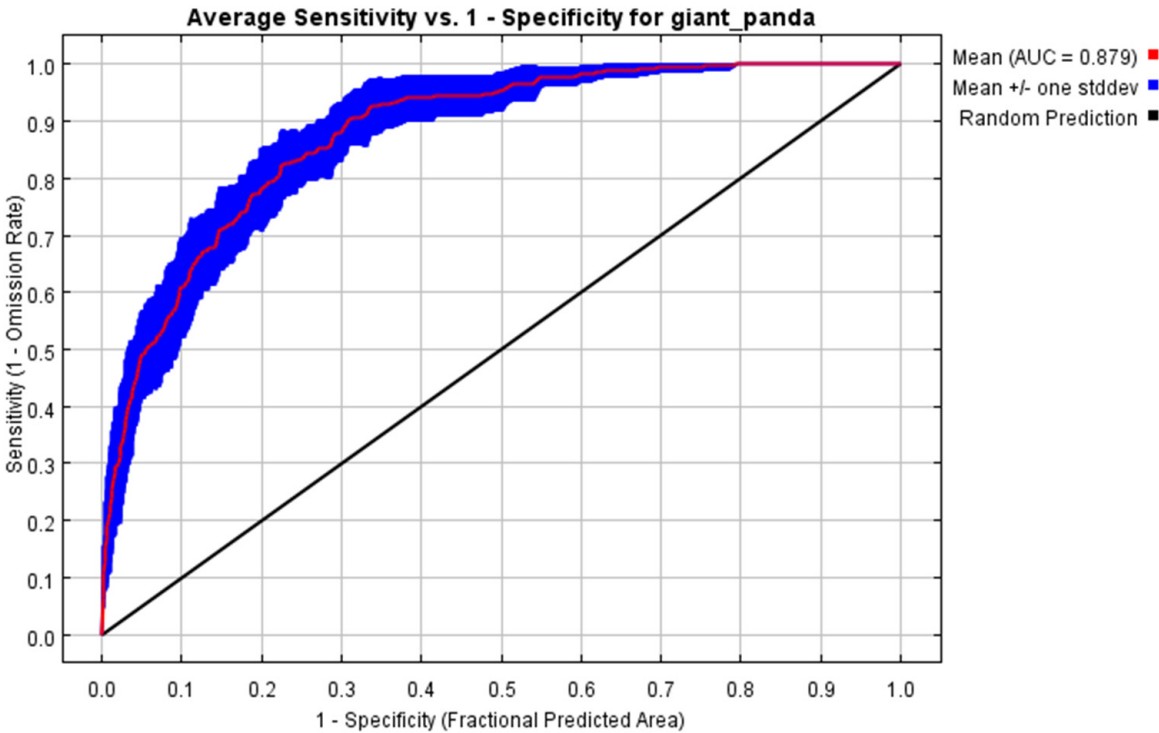

**Figure 2.** Receiver operating characteristic curve of Min Mountain System and area under the curve (AUC).

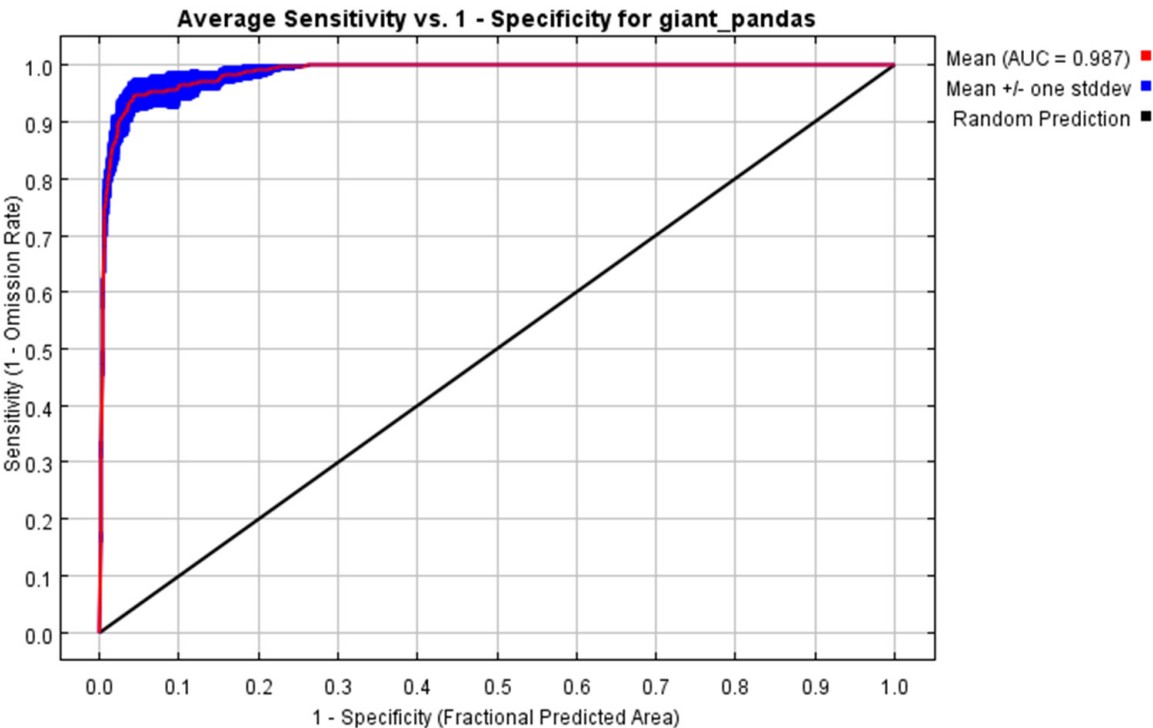

**Figure 3.** Receiver operating characteristic curve of country extent and area under the curve (AUC).

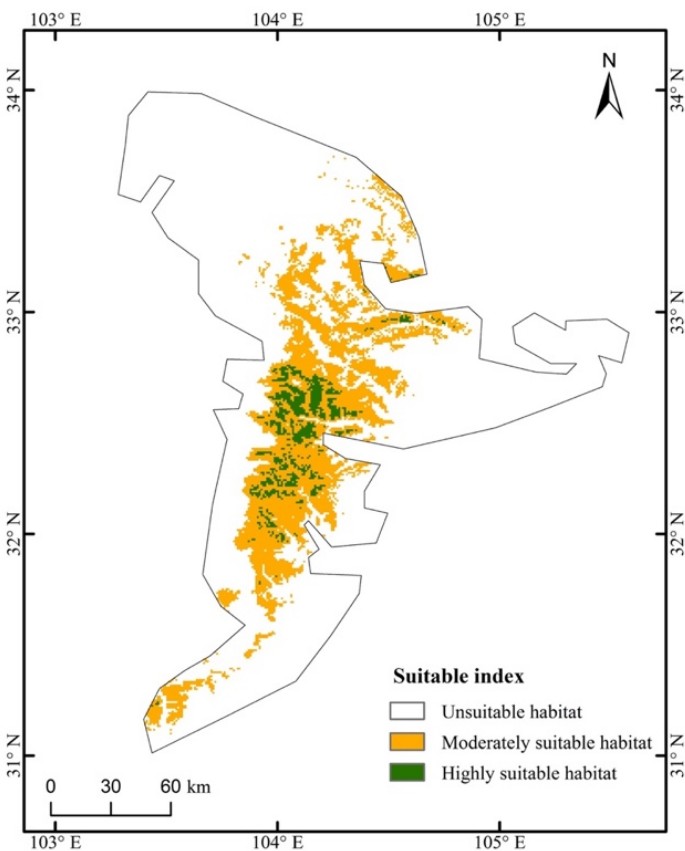

**Figure 4.** The distribution of suitable habitat of wild giant panda in the Min Mountain System obtained by modelling on the Min Mountain extent.

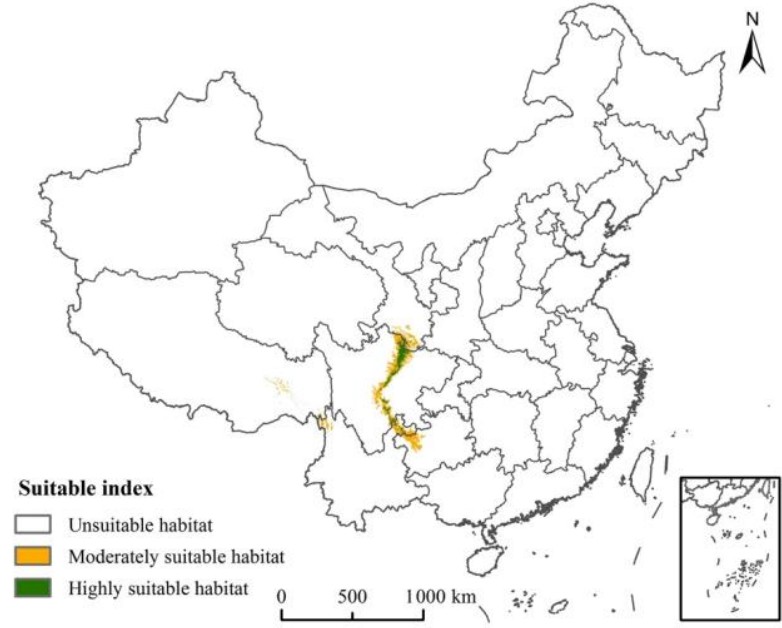

**Figure 5.** Prediction of the distribution probability of wild giant pandas in the Min Mountain System obtained by modelling on the country extent (not clipped).

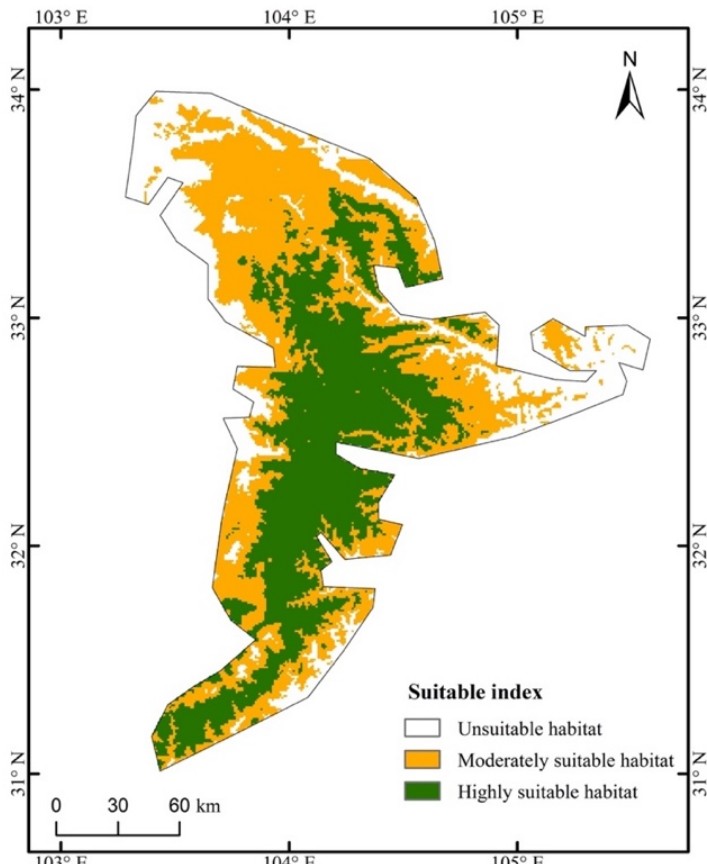

**Figure 6.** Prediction of the distribution probability of wild giant pandas in the Min Mountain System obtained by modelling on the country extent (clipped).

### 3.3. Contribution and Importance of Environmental Predictors

The result of jackknife under the MMS area (Figure 7) and the national range (Figure 8), as well as the percentage contribution rate under two areas (Table 3), are used to test the contribution rate of environmental variables. According to the results of the two methods, under the MMS area, bio15 (coefficient of variation of precipitation seasonality), bio2 (mean diurnal air temperature range), bio14 (precipitation of the driest period) and distance to a stream are the dominant factors affecting the spatial distribution of giant pandas. Under country area, bio12 (annual precipitation), bio3 (isothermality Bio2/Bio7 × 100), bio4 (standard deviation of temperature seasonality) and slope are the dominant factors affecting the geographic distribution of giant pandas.

**Table 3.** Contribution of environmental variables for Wild Giant Panda insignis under the Min Mountain System and Country extent.

| Region | Variables | Contribution (%) | Region | Variables | Contribution (%) |
|---|---|---|---|---|---|
| | Bio2 | 25 | | Bio3 | 23.4 |
| | Bio5 | 0.8 | | Bio4 | 3.6 |
| | Bio14 | 5.4 | | Bio6 | 2.4 |
| | Bio15 | 30.1 | | Bio10 | 1.1 |
| Min Mountain extent | Bio18 | 0.8 | Country extent | Bio12 | 34.4 |
| | Slo | 6 | | Slo | 19.6 |
| | Alt | 9.1 | | Alt | 14.3 |
| | DTC | 2 | | DTC | 0.2 |
| | DTS | 12.9 | | DTS | 0.3 |
| | DTR | 5.3 | | DTR | 0.1 |

Slo = slope; Alt = altitude; DTC = distance to community; DTS= distance to a stream; DTR= distance to road.

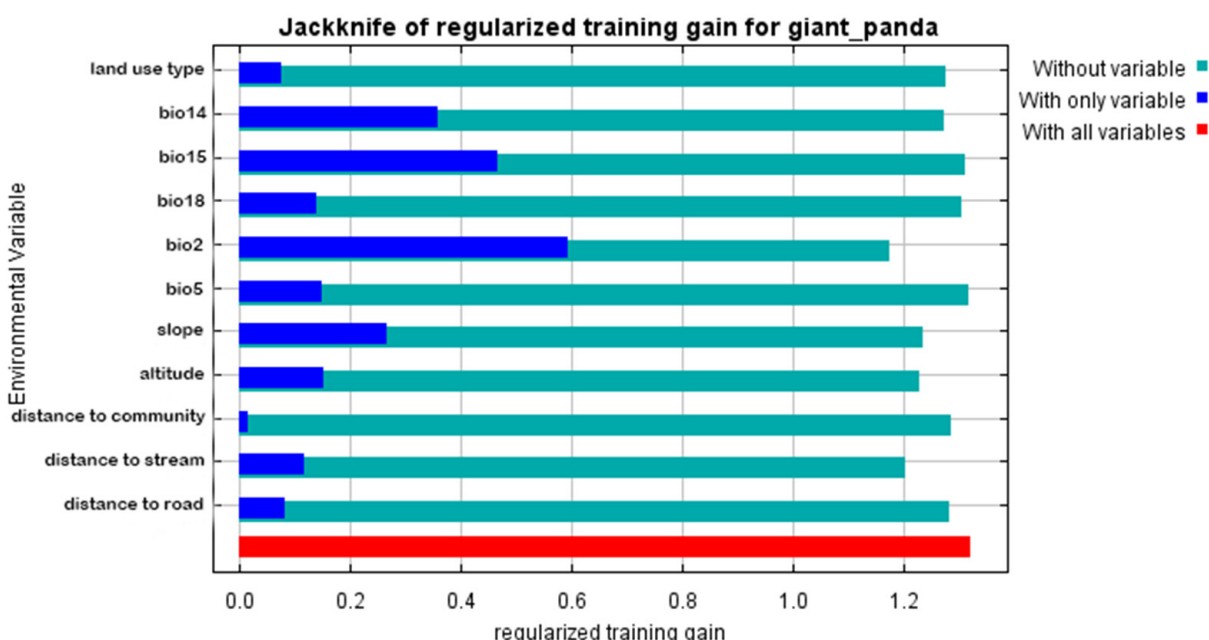

**Figure 7.** Results of jackknife evaluation of the environmental variables concerning regularized training gain for the Min Mountain System.

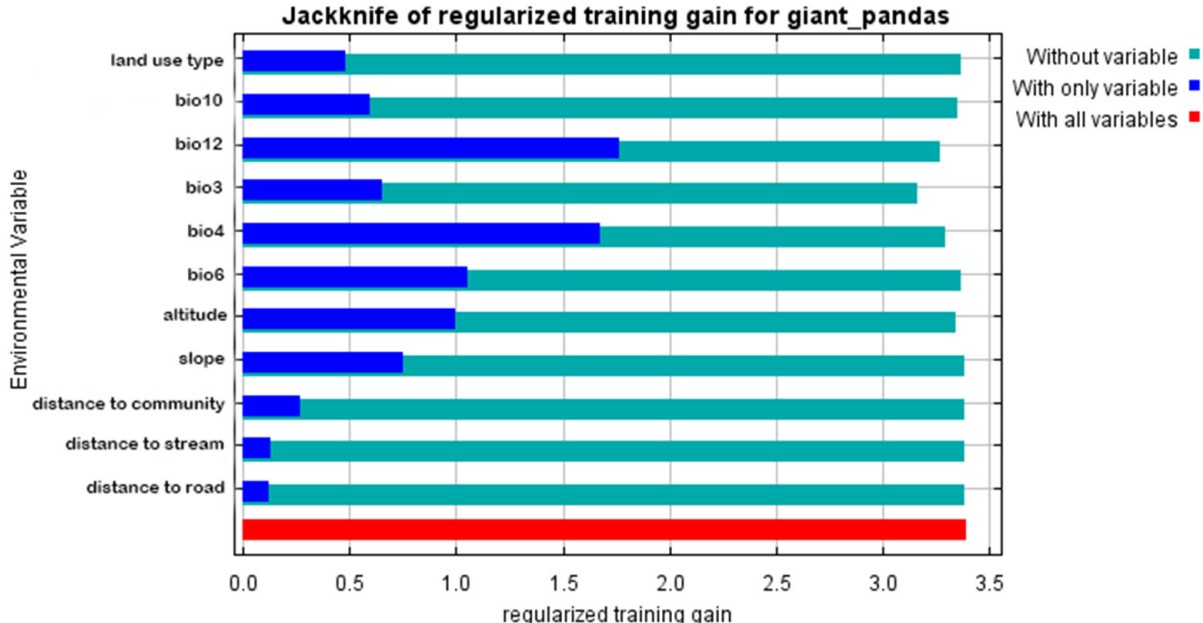

**Figure 8.** Results of jackknife evaluation of the environmental variables concerning regularized training gain for the country extent.

### 3.4. Response Curve

Four factors were selected from the percentage rate and jackknife on the MMS region (Figure 9) and the national region (Figure 10) to conduct the response curve further. The highest point in the figure represents the variable value that is most suitable for the survival of giant pandas. On the one hand, under the MMS area, the results show that the most suitable Mean diurnal air temperature range is about 9.5 °C, and 6–8 mm is the best range of precipitation of the driest month. In addition, 0.15–0.16 (unit: 10 km) is the most suitable

distance from the giant pandas' habitat to a stream, and 65–72 mm is the best range of the value of the precipitation seasonality.

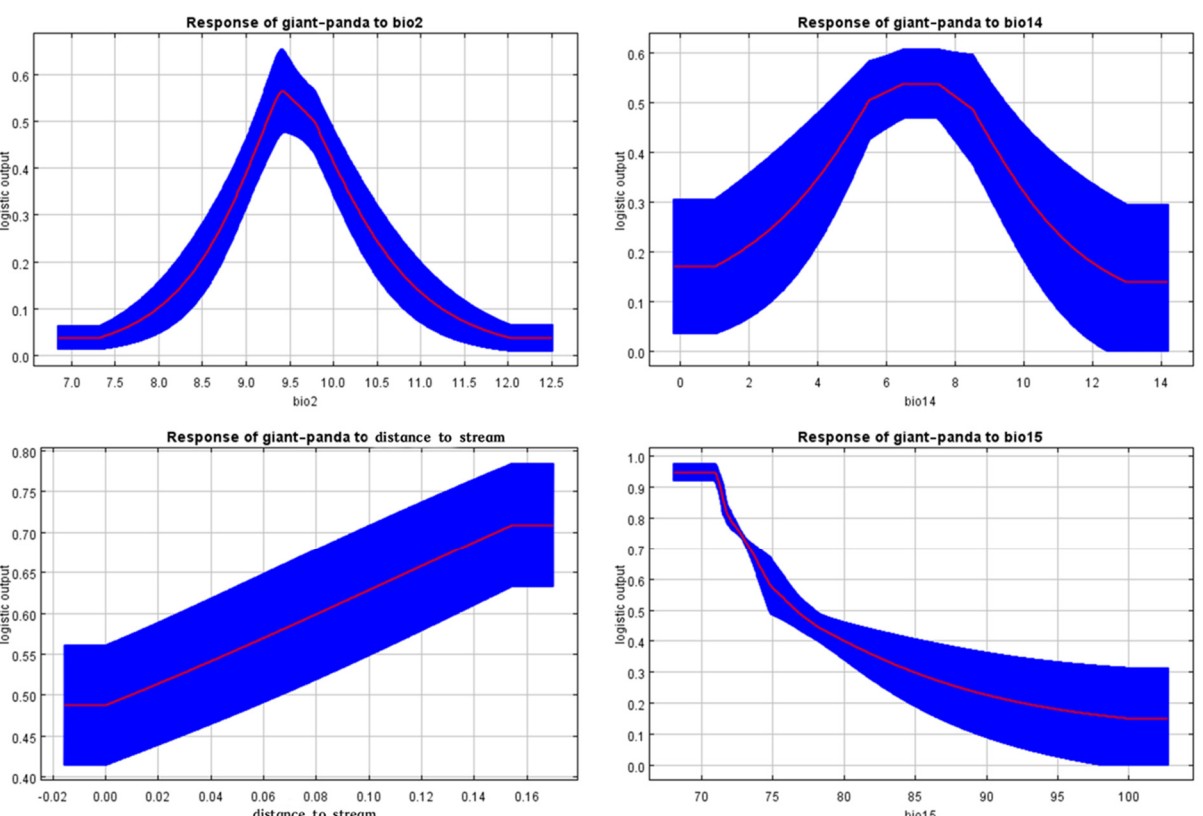

**Figure 9.** The response curve of the Min Mountain extent with the variables bio15, bio2, bio14, and DTS.

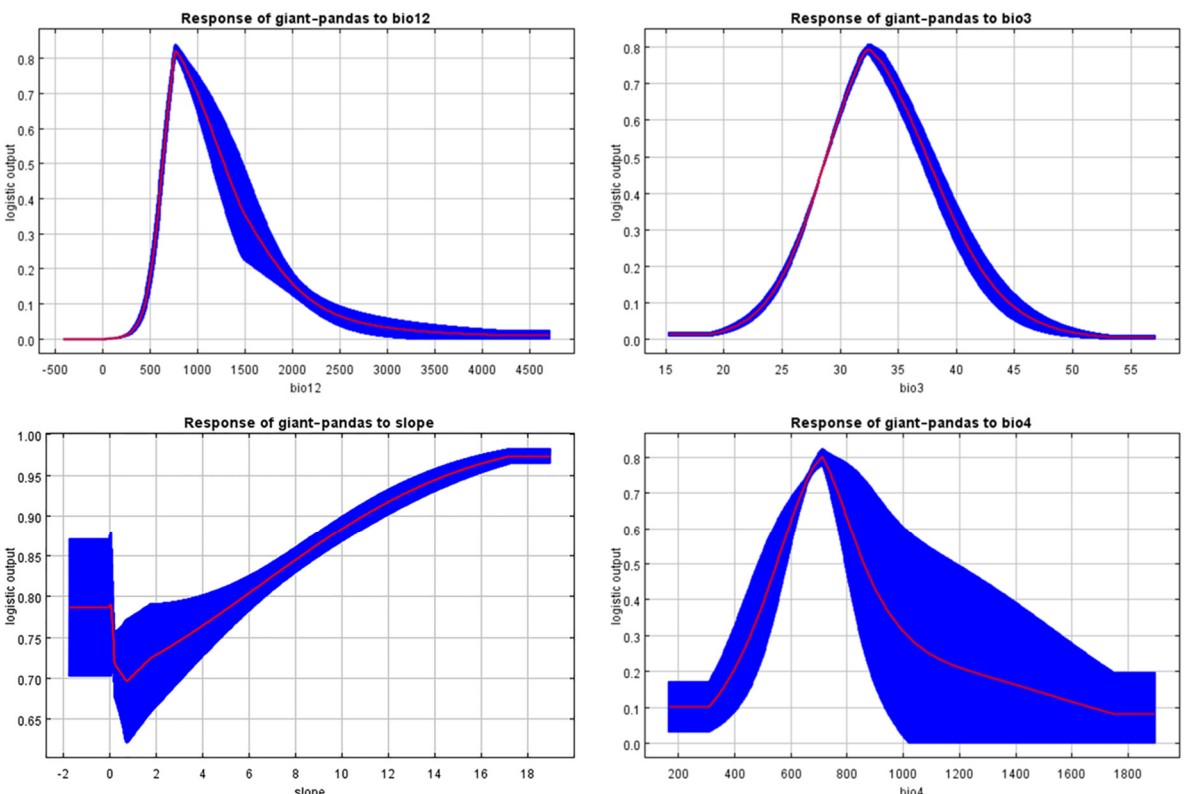

**Figure 10.** The response curve of the country extent with the variable bio12, bio3, bio4, and slope.

On the other hand, under the country extent, an ideal annual precipitation value for giant pandas' survival is about 800 mm, while isothermality (Bio2/Bio7 $\times$ 100) is about 32. The slope's value tends to be ideal at 17 $^\circ$ and above, and the optimal value of the standard deviation of temperature seasonality is about 700.

## 4. Discussion

### 4.1. Model Results Will Be Affected by Errors in the Data Collection

From the process of the data collection and modelling, long data collection time and a lack of environmental and species location data will impact the model's accuracy. Many sites in the MMS areas are underdeveloped, the actual measurement points of the data are sparse, and the observation years of various types of data are not the same, which significantly increases the possibility of data errors [5]. For example, the climate data is from 1970 to 2000, the land type is based on the classification in 2018, and the species distribution data spans several decades. From the process of data collection and acquisition, time inequality is inevitable. Thus, timely updates and data supplements will be necessary to minimize the errors as much as possible and provide better data support for niche modelling.

However, this error has little effect on this study. In terms of the predicted suitable area, the research focus was not to obtain a specific geographical distribution area of giant pandas but to compare the differences in the distribution of the two regions. This error is simultaneously functioning on the two ranges. Therefore, the impact of the time inequality of the data on the research is reduced to some extent.

### 4.2. The Predicted Suitable Areas Are Different at Different Modelling Extents

The suitable habitat maps are quite different at the two spatial extents. However, the highly suitable area in a more extensive range of modelling results almost overlaps with the suitable area (highly suitable area plus moderately suitable area) in the target area modelling results. Thus, the suitable range under larger-extent modelling is more extensive than that under smaller-extent modelling for some reason.

Combined with the accuracy-test results, the model result on a national scale in this study shows a higher prediction accuracy. However, there is an assumption in extreme cases: assuming that the occurrence site data and environmental data are complete and accurate, there is no doubt that the most accurate results will be obtained when the target area is the modelling area. In other words, if the site data and environmental data are very accurate, the model established under the MMS area should be the most accurate.

When the actual results of the model do not match the assumptions, reviewing some phenomena that have not been taken into account in the modelling process can help explain these unexpected results. When the research object is a kind of animal, for example, the giant panda, their mobility and rarity often lead to a lack of data and uncertainty in obtaining distribution sites [5]. Furthermore, giant pandas are sensitive to humans, and many distribution records do not entirely accurately represent the areas where pandas have lived for a long time [5]. Therefore, the lack of distribution sites under the MMS modelling area will underestimate the range of potentially suitable distribution areas [17]. At the same time, modelling at a range more extensive than the target area will overestimate the suitable spatial range. It is easier to obtain more distribution sites and environmental data under a larger spatial scale, which will provide systematic and comprehensive information for establishing a species–environment relationship [17]. Thus, if there is a lack of species distribution sites, appropriately expanding the modelling scope will help offset the possible deviation in modelling at the target area scale.

However, when the research object is a plant, its distribution site will not move, so it is relatively easy to collect. The suitable distribution map from different spatial scales should be identical [17]. Therefore, the degree of movement of the species and the difficulty of data collection is different, and the range that may need to be expanded during modelling is also different.

Based on the above analysis, the MaxEnt model can provide adequate technical support for endangered or difficult-to-track species. For one, the MaxEnt model has low requirements for the number of data. For the other, this study concluded that appropriately expanding the modelling range and clipping out the appropriate scope of the target area may offset the error caused by the lack of distribution sites or environmental variables. The MaxEnt model makes it possible to predict the living conditions of endangered species, and the study of the spatial extent of modelling reduces the possible errors of the model. This provides technical support and guarantees that relevant personnel and agencies can monitor the living conditions of endangered species and intervene if necessary. However, how to expand the scope of the modelling to compensate for the errors caused by the missing sites and variables and to achieve the most accurate results further studies are needed.

### 4.3. Human Interference Factors Need to Be Considered

None of the human interference factors quantified in this study is significant under contribution rate analysis. For one, it may be because human interference shows a gradual weakening trend as the extent increases. Connor et al. (2019) mentioned that human interference, such as the negative impact of roads, is evident at more minor scales but becomes smoother as the total range increases to more than 500 km$^2$ [18]. The target area in this study is much larger than 500 km$^2$, so the influence of human interference is weakened. For the other, the MMS area has more than 30 protected reserves, and there are fewer permanent residents than in other regions [13]. In addition, because fewer roads are connected to the outside in each nature reserve, human disturbance caused by residents' activities and road traffic is not substantial [36].

However, on a more microscopic scale, there are still many human variables that should be considered. For example, the MMS area is the settlement of the Baima Tibetan ethnic minority in China. Grazing horses and yaks have been the traditional production method of this ethnic group for a long time. In recent years, the grazing range has expanded, causing domestic animals and giant pandas to compete for bamboo as a food source, further compressing wild pandas' food and living space [36]. Therefore, grazing activities have become the most influential human disturbance in the MMS [5].

Many empirical studies have mentioned the impact of human interference [8,9]. In order to more accurately predict the habitat area of the giant panda, a standardized quantification system of human interference should be established in the future, or at least combined qualitative analysis to ensure the comprehensiveness of research.

### 4.4. The Contribution Rate of Environmental Variables and Response Curves under Different Modelling Spatial Extents Can Provide New Ideas for Protecting Giant Pandas

Together with geographic extents, the response curve can provide data support for the cross-scale protection of wild giant pandas. It provides references for protection departments of different regions and levels when implementing the protection of giant pandas:

(1) The local managers of each protected area or county in the MMS should pay attention to the management and restriction of human activities on a small scale. Even though the setting scale of this study is too big to identify the influence of human interference, it can be seen from other literature combined with the model result that human interference has actual impacts on the habitat of giant pandas. For example, the effect of increased grazing needs to be taken seriously by local managers to alleviate the compression of the giant pandas' natural habitat.

(2) Managers of relevant departments of the entire MMS area can refer to the environmental contribution rate, and response curve at the MMS extent to take protective measures. For instance, they can monitor the dynamic changes from the factors of 'changes in daily average temperature' and 'precipitation in the driest month' to ensure that the giant pandas' habitat has the correct living conditions. The monitoring and control of these environmental variables can more effectively improve the protection capability within the scope of the Min Mountain System. In addition, monitoring

points can be set up in areas that are within the pandas' suitable distance to a water source to improve wild panda investigation and monitoring efficiency.

(3) From a national macro-scale, national environmental protection organisations and national core ecological protection agencies should adopt macro-monitoring of factors with high contribution rates at large scales to control the fragmentation and degradation of the giant pandas' habitat, for example, monitoring annual precipitation and annual temperature differences.

In future relevant research, if protection or management suggestions need to be made for a certain level of administrative agencies, some data can be collected within their corresponding geographic level. This will help clarify the administrative divisions of giant panda protection agencies and make the division of labor between various administrative groups clearer [10].

## 5. Conclusions

Due to climate change and human disturbance activities, the habitats of endangered wild animals continue to be degraded and fragmented, so their survival is threatened. The research of endangered species can be effectively supported by ENMs, especially the MaxEnt model, which can maintain the stability of model results even when sample data is extremely scarce. At present, the application of the MaxEnt model is mature, but there is still a lack of research on its modelling extent. This study attempts to analyze how changes in the modelling spatial range affect the results of the MaxEnt model, with a view to reducing errors in model research and providing technical support and management suggestions for the sustainable development of giant pandas and their habitats. Taking the MMS region and the MMS region modelled on a national scale as a mutual control group, the main conclusions are as follow: (1) When an endangered species that difficult in data collection of sites is the research object, the modelling range should be appropriately expanded and clipped into the target area to offset errors caused by missing sites; (2) Human factors are generally more significant at a small scale, but human interference at any scale is a factor that cannot be ignored; (3) The contribution rates of environmental variables at different scales are different. Ecological protection departments at different levels should pay attention to changes in the most significant environmental factors at corresponding scales that help control and intervene in potential negative impacts to achieve efficient protection.

The study only suggests that appropriately expanding the modelling scope can help offset the errors caused by the lack of data, but how to accurately offset the errors caused by missing data by changing the extent of modelling requires further research. In addition, the supervision and intervention of species protection agencies are vital to ensure the sustainable development of endangered species, such as giant pandas, and maintain the stability of their habitat. Thus, the idea that administrative agencies at all levels distinguish the management priorities of endangered species through the spatial scope can be further applied to practice to make the effects of supervision and intervention more efficient.

**Author Contributions:** Z.H., conceptualization, data Curation, writing—original draft preparation, validation, formal analysis, visualization; T.P.D., methodology, data curation, supervision, writing—review and editing; L.C., formal analysis, writing—review and editing; A.H., writing—review and editing. All authors have read and agreed to the published version of the manuscript.

**Funding:** This research received no external funding.

**Institutional Review Board Statement:** Not applicable.

**Informed Consent Statement:** Not applicable.

**Data Availability Statement:** Not applicable.

**Acknowledgments:** Thanks to Harriet Dawson (University of Edinburgh) for her review and suggestions for improvement on earlier drafts of this manuscript.

**Conflicts of Interest:** The authors declare no conflict of interest.

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
