# Peer review of "The Effects of the Spatial Extent on Modelling Giant Panda Distributions Using Ecological Niche Models"

_sustainability, doi:10.3390/su132111707_

Round 1

Reviewer 1 Report

This is an interesting manuscript. Assessing how spatial scale influences modelling outputs from ENM is an important topic, particularly as MaxEnt has be shown to perform well with sparse observations, (sometimes even as less as ten records) and that it uses regularization as a form of selecting appropriate predictor variables. My first issue is one for the authors to justify, in quite a broad sense. How does this manuscript fit into the scope of the journal (as opposed to it being submitted to a more techniques based journal such as Remote Sensing, or Remote Sensing in Ecology and Conservation)? What is the sustainability issue being examined here?  

Secondly, there is an issue surrounding the overall aim and objectives of the study. The text in lines 94-109 is very confusing, with the authors using terminology that may confuse the readerships. For a manuscript of this nature, there should really be only one aim, followed by the key objectives (which can be referred to as goals). In my reading, the overall aim is essentially what is presented in the title - to examine how spatial extent influences variation in modelling Giant Panda distribution using ecological niche models. To achieve this aim, the authors quantify anthropogenic disturbance regimes across the known Giant Panda distribution and incorporate these data into new MaxEnt models for the species constructed at two different spatial scales. They then use these data to predict and identify suitable habitat for the species at the two different spatial scales, in order to provide more effective, and scale-dependent targeted conservation measures to sustain viable panda populations. The text presented in lines 100-108 can then be deleted, or at least some of it (e.g. lines 102, 106) be rewoven into the model construction text in the methods section.

Lines 83-85 – there is just not enough background on the status of Giant Panda spatial distribution research, or how, in general, the spatial extent of the study influences the results of MaxEnt models. Firstly there are only three citations here from a limited range of journals and many more are needed to back up the claim that spatial extent influences MaxEnt modelling. Secondly, on line 84, the authors do not mention what results are changed – are they referring to model accuracy or some other output? Thirdly, on line 85 the authors do mention that spatial extent impacts habitat prediction results – but they do not say how.

Lines 112-151 – most of this content is more reminiscent of a review article or a dissertation, but is largely irrelevant for this study and should be deleted. The readership does not need to know about the history of ecological niche definition or modelling.

Lines 172-201 – again most of this text can be deleted as it is irrelevant for this manuscript (again its more  reminiscent of a review article or a dissertation, and represent details not required for the readership). Some of the content on lines 184-201 could (and should) be condensed and merged with lines 83-85 to expand on the background to the use of ENM and giant panda research (see my earlier comments about this).

Lines 225-248 – some of the sentences in this section are written in the first person narrative (present tense) and need to be revised to reflect the third-person past tense narrative (e.g. lines lines 228-229, 235-236 etc).

Lines 262-284 Model construction – generally this is done well but the written text needs to be improved somewhat in order for the readership to evaluate the approach taken (perhaps see Nan et al 2012 page 103 as an example. Firstly, Lines 272-284 –there is a key citation missing. To cite the use of the area under the curve of receiver operating characteristics for model validation and predictive performance, the authors must cite Fielding &Bell (1997). This helps justify the use of AUC versus other measures such as Cohens kappa statistic or true skills statistics. Secondly, there is little justification as to why only two scales are used. And this brings me to my main concern about the analyses. Why just use MMS? Why not reanalyse the data, by repeating the analyses for each region and search for consistency in accuracy and model performance (i.e. use the same approach for the Qinling region, then the Qionglai region etc). The opportunity to do so is there. Alternatively, or additionally, they could explore the addition of each of the six montane regions in subsequent models. Using all montane regions, in a cumulative way, would be a novel approach, and would represent a true multi-scale examination of spatial extent on model accuracy and performance. It may also lead to a greater range of both localised and regional conservation proposals.

Reviewer 2 Report

The manuscript entitled: ‘The effects of the spatial extent on modelling Giant Panda distributions using ecological niche models’ describe the differences in two models using variables in different scales – local and national. The authors used the MaxEnt model to predict the suitability range of giant pandas in the Min Mountain System area. Although the subject of paper is very interesting, is hard to follow since Material and Methods chapter. Moreover, I think that authors mixed in paper the development of models and ecological approach to the subject – what is showed in the goals listed in the Introduction. Authors wrote that they want ‘to provide more practical suggestions for the protection of giant pandas’. It should be the most important conclusion of the research but I didn’t find it. I encourage authors to do some changes, which improve the manuscript.

All my concerns I wrote below:
Abstract: l. 17-18: firstly, both models performed well – what does it mean “well” ? Please be more specific.
Keywords: ecological niche models; the giant panda – these two words are already in the title of manuscript. I suggest use the Latin name of Giant Panda here and skip ‘ecological niche models’.
Introduction: l. 88-93: A summary is not needed here. I suggest skipping that part to go straight to the aims of the research.
l. 96: Please use different word than ‘finally’.
l. 98-109: please skip it – it’s too obvious
l. 113-170: The paper does not focus on ecological niche definition, so those lines should be skipped. Authors should only write what kind of analysis was used. (Moreover, I don’t understand why the authors wrote ’He’ (l. 124) with capital letter. )
l. 172-177: should be added to Introduction or skipped.
l. 179-182: redundant. 
l. 182-191: should be added to introduction or could be used in discussion (l. 187 double space: is Connor). Authors summarized here other author’s research – it is definitely not a place for such things).
L. 171 and 202 – numbering of both chapters is 2.2.
Actually, chapter 2. Materials and Methods should start with text in line 202.
l. 225: you used data on Giant Panda occurrence from what period of time.
Results: describing the results please write 'local/national scale' instead of Figure #. The right number of Figure add in the bracket. It’s very hard to follow when every time quick look for the Figure is needed.
l. 334-335: I don’t know with which part of the manuscript it corresponds.
l. 367: I suggest writing the whole name of Min Mountain System at the beginning of Discussion. 
l. 378: please explain on which result that part of Discussion is based on.
l. 417: I think it is the most important result in the paper and should be more highlighted by authors, especially in conclusions.
The conclusion paragraph should be re-write - must be shorter and summarize the main conclusions from the Discussion Chapter. I don’t understand why each point of Discussion is marked by number.
l. 519: instead of ‘both’ please name them.
l. 528-545: these two points are rather highlights of the paper, not the conclusions.
l. 546: delete ‘According to model data and literature analysis,’ it’s obvious in conclusion section.
l. 550-554: delete.
l. 557-560: delete.
l. 562-595: it should be moved to Discussion chapter.

Round 2

Reviewer 1 Report

I would like to thank the authors for addressing the points I raised in my first review. For their resubmission, I found it difficult to distinguish between revised comments, deleted comments and newly written text as I was unable to see any track changes but was able to see numerous text presented in different colours in their revision. However, it was difficult to determine whether e.g. the green coloured text was previously deleted text, whether red was newly revised etc. Now this may simply be an issue with the digital format I can see, rather than the editorial changes made by the authors per se. So this second review does come with that caveat – changes may have been made but I am unable to distinguish between them on the resubmitted/revised manuscript. So I’ll do my best here, but I apologise in advance to the authors if I make a suggestion that has already been addressed.

Sustainability – I thank the authors for outlining their rationale on submitting this manuscript to sustainability in the response document. However, to me the main sustainability issue they are addressing is much more simple – that sustaining viable globally threatened species populations requires suitable evidence-based conservation strategies from meaningful scales. And that’s the main content lacking in the Introduction, a much broader acknowledgement that Panda conservation requires sustainable evidence-based conservation strategies. This really does need to be spelled out in clearer, more simpler terms.   

Lines 42-72: Although I can see that the authors are outlining the general background to the status of Giant Panda populations here, the overall standard of the written English grammar throughout this section of the Introduction needs to be improved substantially. In addition, there are numerous editorial errors throughout (see lines 45-45, lines 65-67 etc). I would encourage the authors to spend some time going over the whole manuscript line by line, perhaps with the help of other English speakers to improve the grammar but also ensure there are no additional editorial errors.

Lines  83-86: I don’t see the need for this text and it can be deleted.

Lines 90-91: I would revise this sentence to “ Presently, there is an urgent need for quantifying how different Giant Panda populations are predicted to respond to different climate change scenarios.

Lines 96-106:  I suggest revising this paragraph to the following text: “Few studies have examined the influence of spatial scale on the predictions generated by MaxEnt modelling. Of these, some studies have shown that different scales have minimal influence on model predictions (e.g. Zhuang et al., 2018), whereas other studies have concluded that variation in scale-dependent effects are influenced by environmental variable selection (e.g. Zhen 2018; Connor 2019). To our knowledge, there remain few, if any, concrete recommendations on how to deal with these issues in order to improve MaxEnt modelling.

Lines 113-116: This simply repeats the Aim and objectives in the following paragraph. I’m not sure why its highlighted in yellow colour either. I suggest deleting this.

Lines 118-141: This is not much different from the original submission. I refer to the comments I made in my first review, for a more concise outlining of the aim and objectives, and encourage the authors to use similar wording in their revision. In my reading, the overall aim is essentially what is presented in the title - to examine how spatial extent influences variation in modelling Giant Panda distribution using ecological niche models. To achieve this aim, the authors quantify anthropogenic disturbance regimes across the known Giant Panda distribution and incorporate these data into new MaxEnt models for the species constructed at different spatial scales. They then use these data to predict and identify suitable habitat for the species at two different spatial scales, in order to provide more effective, and scale-dependent targeted conservation measures to sustain viable panda populations.

Lines 143-180: This text is presented in green colouration and I’m hoping this is the text that has been deleted from the original submission? Apologies again to the authors if this is the case due to the digital format I can see, but I am more used to seeing track changes in a revised document, rather than repositioned text presented in different colours.

Lines 191-193 – this text needs to be revised – I suggest the following: “MaxEnt models exhibit little sensitivity or changes to model accuracy with significant changes in data quantity, although processing times tend to be accelerated, model accuracy is higher when models are based on presence-only data (reference).

Lines 193-200: As per my comment earlier about green coloured text, I’m assuming this should be deleted from the manuscript?

Lines 203-241: I don’t think any of this text is relevant for the manuscript – again is far more suited to a review article or a dissertation thesis – this can all be deleted.

Lines 300: Change “is selected” to “was selected”

Lines 300-304: Suggest revising this to “We selected the national extent as the control group, which contained all the known global distributional data for Giant Panda. Comparing of the model result under the MMS region and the nationwide extent (subsequently clipped to the MMS range) enabled us to evaluate variation in predicted suitable habitat areas.

Lines 327-336: I suggest revising this: “Species distribution data were derived from two sources; firstly the Global Biodiversity Information Faculty [26] and National Specimen Information Infrastructure [27]; and secondly from all relevant peer-review literature records of giant panda's occurrence. This resulted in 260 occurrence sites. Google Earth was used to filter the distribution information to gain the relevant latitude and longitude coordinates, as well as helping to remove ‘double records’ and locations with unknown geographic coordinates. This resulted in 89 occurrence records of which, 65 were from the MMS.”

Lines 338-343: This is rather poorly written and on the whole, not necessary. I suggest removing this text from the manuscript.

Lines 343-345: Revise this text – I suggest the following: “All climate and environmental data was obtained from WORLDCLIM [29] – a global database based on meteorological information recorded by weather stations worldwide from 1970 to 2000 (Table 2).

Lines 368-369: Change this line of text to the following: “Distribution and environmental data were imported into MaxEnt.”

Line 378: Are you referring to the Jackknife text which examines the importance of each environmental variable (i.e. its weight)?.

Lines 378-394: This text needs to be revised. I suggest reviewing the text from Nan, L., Chen-Xi, J., Lloyd, H. & Yue-Hua, S. (2012). Species-specific habitat fragmentation assessment, considering ecological niche requirements and dispersal capability. Biological Conservation 152: 102-109. Page 103. This will help you produce a much more concise, revised text.

Lines 402: What do you mean by authentic? Do you mean accurate?

Lines 398-403: Revise this text to the following: “AUC values of the MMS region and the national extent were 0.879 and 0.987, respectively, suggesting that the MaxEnt models for both scales performed well, and were both highly authentic, with the national extent scaled model being more accurate than the MMS regional model.”  

Line 410: Revise the subheading to “Distribution of predicted suitable areas for Giant Panda”

Line 434: Revise this to “Contribution and importance of environmental predictors”

Lines 435-436: Do you mean the results of the Jacknife test?

Lines 435-444: This text again needs cleaning up – a few grammatical and editorial errors here.

Line 463: Change ‘are selected’ to ‘were selected’

Line 473: What are you referring to here? What can be more suitable? Do you mean that predicted suitable habitat areas generally have an annual precipitation of 800mm, based on the national extent modelling scale?

Line 474: The most suitable suit doesn’t really make much sense – suggest revising this term

Lines 486-297: This is not needed – you are simply repeating the aim and objectives along with the methodology. Delete this and get straight to the most important findings of your research.

Discussion: Editorially this is not well presented, with large conspicuous indentations on the pages and irregular numbering of subheadings, almost in a bullet point format. Overall it is far too long and I encourage the authors to work on reducing this, cutting out a lot of the repeated content, and stress the compare and contrast element with other studies. In addition, the emphasis on sustainable, viable panda populations needs to be made much more clearer in a greatly condensed text.

Reviewer 2 Report

The manuscript was corrected according to suggestions. I have no other comments.

Author Response

Thank you for your previous detailed suggestions, which has further improved the article.

Round 3

Reviewer 1 Report

I thank the authors for working on my previous comments. There are still many English grammar issues that remain and I would like to encourage the authors to reach out to another academic who can perhaps help to refine the English language and writing style. Analytically, the manuscript is ready, but the English language and grammar needs more revision. 

Author Response

Some of the presentation errors and grammatical errors in the article have been corrected by a native English-speaking colleague.